# Growth and DNA Methylation Alteration in Rice (*Oryza sativa* L.) in Response to Ozone Stress

**DOI:** 10.3390/genes14101888

**Published:** 2023-09-28

**Authors:** Hongyan Wang, Long Wang, Mengke Yang, Ning Zhang, Jiazhen Li, Yuqian Wang, Yue Wang, Xuewen Wang, Yanan Ruan, Sheng Xu

**Affiliations:** 1Laboratory of Plant Epigenetics and Evolution, School of Life Sciences, Liaoning University, Shenyang 110036, China; hongyan2003@126.com (H.W.); wanglong5898@163.com (L.W.); yangmengke11111@163.com (M.Y.); zhangning6123@126.com (N.Z.); lijiazhen622@163.com (J.L.); wangyuqian0423@163.com (Y.W.); wangyue122097@163.com (Y.W.); 2Academy of Agricultural and Forestry Sciences, Qinghai University, Xining 810016, China; 3Department of Genetics, University of Georgia, Athens, GA 30602, USA; xwvan@163.com; 4CAS Key Laboratory of Forest Ecology and Management, Institute of Applied Ecology, Shenyang 110016, China

**Keywords:** ozone, DNA methylation, methylation-sensitive amplified polymorphism (MSAP), rice (*Oryza sativa* L.)

## Abstract

With the development of urban industrialization, the increasing ozone concentration (O_3_) at ground level stresses on the survival of plants. Plants have to adapt to ozone stress. DNA methylation is crucial for a rapid response to abiotic stress in plants. Little information is known regarding the epigenetic response of DNA methylation of plants to O_3_ stress. This study is designed to explore the epigenetic mechanism and identify a possible core modification of DNA methylation or genes in the plant, in response to O_3_ stress. We investigated the agronomic traits and genome-wide DNA methylation variations of the Japonica rice cultivar Nipponbare in response to O_3_ stress at three high concentrations (80, 160, and 200 nmol·mol^−1^), simulated using open-top chambers (OTC). The flag leaf length, panicle length, and hundred-grain weight of rice showed beneficial effects at 80 nmol·mol^−1^ O_3_ and an inhibitory effect at both 160 and 200 nmol·mol^−1^ O_3._ The methylation-sensitive amplified polymorphism results showed that the O_3_-induced genome-wide methylation alterations account for 14.72–15.18% at three different concentrations. Our results demonstrated that methylation and demethylation alteration sites were activated throughout the O_3_ stress, mainly at CNG sites. By recovering and sequencing bands with methylation alteration, ten stress-related differentially amplified sequences, widely present on different chromosomes, were obtained. Our findings show that DNA methylation may be an active and rapid epigenetic response to ozone stress. These results can provide us with a theoretical basis and a reference to look for more hereditary information about the molecular mechanism of plant resistance to O_3_ pollution.

## 1. Introduction

Ozone (O_3_), one of the greenhouse gases in the atmosphere, is a colorless gas with a pungent odor and strong oxidability [1,2]. Its concentration has increased in the troposphere with urban industrialization [3]. Some studies speculate a continuous increase in O_3_ in the troposphere at a 1–2% rate per year from 2015 to 2050, which will rise to 40–60% by 2100 [4]. This may hinder plants’ growth, reducing yields under O_3_ stress, especially for crops [5,6,7]. O_3_, with its strong oxidability, can enter plant tissues through stomata and be converted into reactive oxygen species (ROS), producing oxidative stress in plants. As a result, less carbon dioxide enters the leaves due to the decreased plant stomatal conductance, which inhibits carbon assimilation, and reduces the net photosynthetic rate of plants. It also damages cell membranes, lipids, and enzymes, hampering plant growth and development [8,9]. An assessment of O_3_ pollution in East Asia showed the highest relative yield losses were 42.2%, 32.6%, and 8.6% for rice, wheat, and maize, respectively, in China, resulting in financial losses of $63 billion in East Asia [10]. In addition, the economic loss of crops caused by O_3_ is expected to be up to $26 billion per year by 2100, posing a significant financial challenge globally [11].

Plants must adapt to the constant change in their living environment [12]. Many studies have shown that epigenetic mechanisms are vital for plant responses to abiotic stress [13,14,15]. DNA methylation, which is the covalent addition of the methyl group at the 5-carbon of the cytosine ring resulting in 5-methylcytosine (5-mC), is regarded as one of the main epigenetic mechanisms that can regulate ontogeny and phylogeny, as well as the primary mechanism for genome defense, stress response, and parental imprinting [16,17]. In plants, cytosine methylation can occur in all contexts (CG, CHG, and CHH, where H is A, T, or C) [18]. Plants can regulate the expression of stress-responsive genes via dynamic changes in DNA methylation, enhancing their adaptability to the environment [16,19]. Previous methylation-sensitive amplified polymorphism (MSAP) analysis revealed that oilseed rape under salt stress was modified by extensive demethylation, enabling its stress tolerance [20]. It was also revealed that *Arabidopsis thaliana* regulated the dynamic changes in DNA methylation to respond to pathogen stress [21]. In addition, MSAP analysis showed that decreased methylation levels in 5-azaC-induced kenaf seedlings played an essential role in mitigating the damage caused by salt stress [22]. These results suggested that variations in DNA methylation levels and patterns are associated with abiotic stress responses in plants, which vary among plant species.

The increasing O_3_ concentration in the troposphere is the third-most potent anthropogenic greenhouse gas [11]. As a vital food crop planted and grown in areas and seasons subject to O_3_ pollution, rice (*Oryza sativa* L.) is one of the most severely affected crops, with a reduction in its output of up to 42.2% caused by O_3_ pollution in China alone [10,11]. The cause of the reduced production is now widely believed to be O_3_ entering into the tissues through leaf stomata and producing reactive oxide species (ROS), such as hydrogen peroxide, followed by oxygenolysis, which damages plant growth and development [23]. In response to the ROS produced by O3 stress, plants must establish important active mechanisms to respond to this abiotic stress, especially at the molecular level. ROS is well-known to induce epigenetic remodeling in biotic or abiotic stress [24,25], but little information is known on the epigenetic response of plants to O_3_ stress. Therefore, studying epigenetic changes, such as DNA methylation variations under O_3_ stress in rice, is particularly important.

In this study, we attempt to explore the epigenetic mechanism and identify possible core modification of DNA methylation in rice in response to different concentrations of O_3_ stress using the MSAP technology. We hypothesized that: (1) DNA methylation is crucial for the rapid response to O_3_ stress in rice; (2) different concentrations of O_3_ stress could cause different phenotypic and agronomic changes, which correspond to DNA methylation alterations in rice; (3) given that demethylation is associated with gene expression activation, O_3_ stress could induce demethylation alteration in rice. Therefore, it will increase the synthesis of osmotic adjustment substances and the activity of antioxidant systems to balance the damage of ROS. To our knowledge, this is the first research to investigate the relationship between DNA methylation variation and O_3_ stress. These results can provide a theoretical basis and reference for further examining plant resistance mechanisms to O_3_ stress and for breeding new O_3_-resistant rice varieties.

## 2. Materials and Methods

### 2.1. Plant Materials and O_3_ Stress Treatment

The Japonica variety Nipponbare (*Oryza Sativa* Japonica. cv. Nipponbare) was used in this study. The seeds were sterilized in 75% ethanol (*v*/*v*) before germination in the dark at 25 °C for 3 days. Germinated seeds were planted in pots containing peat, vermiculite, and perlite (10:1:1, V:V:V). Rice seedlings were grown in an incubator at 28 °C/22 °C(light/dark) until the fourth leaves were fully expanded. All the plants, during the same vegetative growth phase, were moved to the open-top chambers (OTCs) and experienced fumigation under the different O_3_ concentrations, including control (air concentration, approximately 40 nmol·mol^−1^), 80 nmol·mol^−1^, 160 nmol·mol^−1^, and 200 nmol·mol^−1^ for 5 d, 10 d, 15 d, 20 d, and 30 d, respectively. Three OTCs (replicates) were used for one treatment. The OTC design information and O_3_ control systems are described in Xu et al. [26]. During the experiment, leaf tissues from the different treatments were collected, frozen in liquid nitrogen, and stored at −80 °C. Three replicates at each time point were used for DNA methylation analysis.

### 2.2. Measurement and Statistics of Agronomic Traits of Rice

The agronomic traits (panicle length, hundred-grain weight, flag leaf length, and flag leaf width) of rice were measured after 30 days of O_3_ exposure at each concentration. Then, the data were processed and counted with SPSS 25.0 and R software. The *t*-test was used for significant difference analysis between the different treatment and control groups (0.01 < *p* < 0.05, denoted by *; *p* < 0.01, denoted by **).

### 2.3. MSAP Analysis on Genome of Rice Leaf

Rice genomic DNA was extracted by modified CTAB and analyzed by MSAP for methylation levels, patterns, and sites [27]. Information on the digestion adaptors and primers is shown in Appendix A. The methylation level (MSAP%) was calculated according to the following equation:MSAP (%) = [(II + III + IV)/(I + II + III + IV)] × 100%(1)

### 2.4. Isolation and Identification of Specific Variant Bands

Differentially amplified fragments were excised from the gel and incubated at 37 °C for 8 h for further studies. The supernatant was recovered by centrifugation and used for re-amplification, followed by ligation and transformation using DH-5α competent cells (Beijing Dingguo Changsheng Biotechnology Co., Ltd., Beijing, China) and the pUM19-T vector (Beijing Dingguo Changsheng Biotechnology Co., Ltd., Beijing, China). The recombinant DNA fragments were cloned and sequenced for BLAST homology probing based on sequence features.

### 2.5. DNA Sequence Alignment and Homology Probing

The target variant sequences were input into EnsemblPlants (http://plants.ensembl.org/Setaria_italica/Tools/Blast?db=core, accessed on 6 May 2022), Phytozome (https://phytozome.jgi.doe.gov, accessed on 6 May 2022), and the NCBI website (http://blast.ncbi.nlm.nih.gov/Blast.cgi, accessed on 6 May 2022) for query and comparison. Then, they were explored and localized in the whole rice genome by Blast-N search. Finally, these sequences were queried and compared on the NCBI website (http://blast.ncbi.nlm.nih.gov/Blast.cgi, accessed on 6 May 2022) and subjected to homology probing and analysis by Blast X search.

## 3. Results

### 3.1. Damage to Rice Leaves Caused by O_3_ Stress

The most direct phenotypic damage caused by O_3_ stress to plants is reflected in their leaves [8]. Observation of rice leaves exposed to the different O_3_ concentrations (Figure 1) showed no apparent change in the color of leaves compared with the control group following O_3_ treatment at a concentration of 80 nmol·mol^−1^, but a significant difference was observed in groups subjected to O_3_ treatment at 160 nmol·mol^−1^ and 200 nmol·mol^−1^. Specifically, chlorosis occurred in rice leaves after 20 days of treatment at 160 nmol·mol^−1^ O_3_, with small yellow necrotic lesions on their surface; aggravated chlorosis was observed at 200 nmol·mol^−1^ O_3_, with deep yellow and larger necrotic lesions. Compared with 160 nmol·mol^−1^ O_3_, the 30-day treatment of 200 nmol·mol^−1^ O_3_ caused a more significant increase in the rice damage where the leaf color was close to yellow and showed a large area of brown necrotic lesions.

### 3.2. Analysis of Agronomic Traits of Rice under Ozone Stress Treatment

We further measured and counted the treated rice’s agronomic traits (flag leaf length, flag leaf width, panicle length, and hundred-grain weight) during the vegetative and reproductive growth phases (Figure 2). The results showed a significant difference between high O_3_ concentration and the control in the agronomic traits of rice (*p* < 0.05) except for flag leaf width. A considerably long panicle length and flag leaf length were observed at 80 nmol·mol^−1^ O_3_ compared with the control group; the flag leaf length was significantly inhibited at 160 nmol·mol^−1^ O_3_ (*p* < 0.05), and the panicle length and hundred-grain weight were significantly inhibited at 200 nmol·mol^−1^ O_3_ (*p* < 0.01)_._ Despite finding no significant difference in the agronomic trait of flag leaf width compared with the control group, numerically, we found that O_3_ stimulated its growth at 80 nmol·mol^−1^ O_3_ and inhibited its growth at both 160 nmol·mol^−1^ and 200 nmol·mol^−1^ O_3_ compared with the control group. 

### 3.3. Extensive Methylation Alteration in Rice Genomic DNA Induced by Ozone Stress

To analyze the extent of the DNA methylation alteration of rice under O_3_ stress, the MSAP technique was used to detect the methylation variations of the genome-wide DNA of rice, following O_3_ stress treatment at different concentrations. There were 1467–1481 clear methylated bands amplified, with 1515 explicit sites (Table 1, Figure 3). The comparative analysis revealed that extensive DNA methylation variations were induced rapidly after 5 days of O_3_ treatment at all concentrations in rice. According to different digestion combinations, the bands were categorized into four types: non-methylated sites (type I bands) were 1285–1292, accounting for 84.69–85.54% of the total sites; hemimethylated sites (type II bands) were 53–66, accounting for 3.50–4.36% of the total sites; and fully methylated sites (type III and IV bands) were 157–174, accounting for 10.36–11.49% of the total sites. There were 223–230 polymorphic methylated bands (type II + III + IV bands), accounting for 14.72–15.18% of the total number, indicating that O_3_ stress could induce genome-wide demethylation alteration in rice. Full methylation was higher than hemimethylation, and the methylation rate of the CG sites was higher than that of the other two sites. The overall DNA methylation at the early stage (≤10 d) of O_3_ stress at three concentrations showed no significant change, while a significant reduction was seen with the increase in treatment time. These showed that DNA methylation could rapidly respond to O_3_ stress in rice, and that O_3_ stress at different concentrations caused extensive methylation variations genome-widely, at both fully and hemimethylated sites. In addition, we found significantly fewer type II bands under O_3_ stress at 80 nmol·mol^−1^ than at 160 nmol·mol^−1^ and 200 nmol·mol^−1^, and the opposite was true for type IV bands.

To further compare and analyze whether O_3_ stress could induce hypermethylation or hypomethylation alteration in the whole rice genome, the amplification results of MSAP were divided into three categories (Figure 4), including no methylation change, cytosine demethylation, and cytosine hypermethylation. The findings showed that O_3_ stress at different concentrations could induce extensive demethylation and hypermethylation variations in the rice genome. Among them, the demethylation sites accounted for 0.73–2.18%. Specifically, demethylation sites increased from 0.73% at 5 days to 2.18% at 30 days at 80 nmol·mol^−1^ O_3_, and from 1.12% to 1.98% at 160 nmol·mol^−1^ O_3._ However, it decreased from 1.25% to 0.99% at 200 nmol·mol^−1^ O_3,_ showing a rise–fall trend. It showed that with the increase in treatment time, O_3_ stress at 80 nmol·mol^−1^ and 160 nmol·mol^−1^ were more likely to induce demethylation compared with that at 200 nmol·mol^−1^. In addition, hypermethylation induced by O_3_ stress at different concentrations accounted for 0.79–1.06% of the overall methylation sites in rice, as shown by the increase in hypermethylation sites from 0.86% at 5 days to 1.06% at 30 days at 80 nmol·mol^−1^ O_3_, the lack of increase from 0.86% at both 5 and 30 days at 160 nmol·mol^−1^ O_3_, and the decrease from 0.92% at 5 days to 0.79% at 30 days at 200 nmol·mol^−1^ O_3_. Overall, O_3_ in three concentrations induced methylation and demethylation variations in rice samples compared with the control group, with a higher level of demethylation than methylation (Figure 4). In addition, it was interesting that the DNA methylation alteration was more significant in rice at 80 nmol·mol^−1^ O_3,_ and it was speculated that this DNA methylation and demethylation might regulate more functional gene expression in respond to O_3_ stress.

### 3.4. Analysis of Overall DNA Methylation and Demethylation Sites in Rice under Ozone Stress

The hypermethylation and hypomethylation variation sites were counted to further elaborate on the patterns of methylation alteration in the rice genomic DNA after O_3_ stress (Figure 4). The results showed that the CNG, CG, and CG/CNG sites with hypomethylation accounted for 0.594–1.584%, 0.132–0.396%, and 0.066–1.98% of the total sites, respectively (Figure 4A). The hypomethylation alteration in rice genomic DNA was found at the CNG and CG sites after 5 days of O_3_ stress treatment at all concentrations and at the new CG/CNG sites after 10 days at 80 nmol·mol^−1^ O_3_. Furthermore, increased hypomethylation alterations occurred at the CNG and CG/CNG sites of rice genomic DNA at both 80 nmol·mol^−1^ and 160 nmol·mol^−1^ O_3_; in contrast, a rise–fall trend was found in the hypomethylation alterations at the three sites at 200 nmol·mol^−1^ O_3_.

Analysis of the hypermethylation variation sites suggested that hypermethylation that occurred at CNG, CG, and CG/CNG sites accounted for 0.396–0.858%, 0.066–0.132%, and 0.066–0.33% of the total sites, respectively (Figure 4B). The hypermethylation alteration occurred only at the CNG sites at 5 days at 80 nmol·mol^−1^ O_3_, while the methylation at CG and CG/CNG sites was activated with increasing O_3_ concentration. During O_3_ stress, hypermethylation alteration at the CG/CNG sites showed a rise–fall trend at 80 nmol·mol^−1^ O_3_, followed by another continuous increase in the hypermethylation alteration at the CG/CNG sites at both 160 nmol·mol^−1^ and 200 nmol·mol^−1^ O_3_. With the increased treatment time, the hypermethylation alteration of rice genomic DNA could be observed at the CNG, CG, and CG/CNG sites at 30 days at all O_3_ concentrations. In addition, hypermethylation at the CNG site occurred at all three O_3_ concentrations.

Most hypermethylation and hypomethylation alterations tended to occur at CNG sites with different alteration patterns. Moreover, the increase in stress activated more hypomethylation and hypermethylation sites to respond to this O_3_ stress. Thus, this extensive methylation modification in plants may be an important mechanism in their response to ozone stress.

### 3.5. Differential Methylation Sequence Analysis

To determine the characteristic of DNA methylation variation sequences under O_3_ stress, 153 bands were cloned and sequenced. A homology comparison suggested that only 9 sequences had high homology (Table 2). Eight of these nine sequences were demethylated. The 9 functional genes were widely distributed on the rice genome, with more variant sequences on chromosome 11. According to the BLAST results, these sequences were homologous to the genes involved in many processes, including hormone regulation, protein transport, plant development, and transposon activation.

## 4. Discussion

Studies have demonstrated that abiotic stress can induce changes in genome-wide cytosine methylation [20,21], making the correlation between DNA methylation and abiotic stress tolerance a trending topic in plant research [28,29,30,31]. DNA methylation is critical in the plant’s response to adversity and stress. It can regulate gene expression at the epigenetic level by interacting with transcription factors or altering chromosome structure. The MSAP technique was used to assess the extent and pattern of DNA methylation in response to O_3_ stress. Overall, this study found a link between O_3_ stress and methylation levels. The results showed that DNA methylation levels were mainly reflected in changes in both full methylation and hemimethylation levels under O_3_ stress, with the former being higher than the latter. In addition, the overall DNA methylation levels showed no significant change at the early stage (≤10 d) of O_3_ stress at three concentrations but showed a considerable decrease with increased treatment time. We also found that overall demethylation alteration was detected under the three concentrations of O_3_ stress, mainly at the CNG site, which was in line with previous findings that the CNG sites are more vulnerable to abiotic stresses [32].

The analysis of rice’s phenotypic and agronomic traits suggested no significant damage in the leaves, and their vegetative and reproductive growth was promoted at 80 nmol·mol^−1^ O_3_ compared with the control group. However, the leaf surface showed damage and necrotic lesions at 160 nmol·mol^−1^ and 200 nmol·mol^−1^ O_3_. The higher the O_3_ concentration, the more severe the damage to the leaf surface, with varying degrees of the inhibition of agronomic traits (Figure 2). This might be explained by more stress-related genes being mobilized by extensive demethylation modification in rice at an O_3_ concentration of 80 nmol·mol^−1^ to adapt to O_3_ stress. Research has reported that purple acid phosphatase (MS99), zinc finger protein family genes (MS111), and L-isoaspartate methyltransferase (MS96) are involved in ROS scavenging, synthesis of metabolites, and the reduction in reactive oxygen species accumulation [33,34]. In addition, phosphoenolpyruvate carboxykinase (MS14, MS78), membrane HPP family proteins (MS93), KH structural domain functional proteins (MS114, MS120), and transposons (MS107) may also play a role in fruit ripening, stomatal opening, and material transport as well as in DNA damage repair [35,36].

O_3_ stress disrupts the normal physiological homeostasis of plants, which is followed by a massive accumulation of reactive oxygen species, which produce oxidative stress on plants and consequently inhibit plant growth and development [8,9]. In response to abiotic stress, plants mobilize their complex mechanisms, which is encouraged by altered DNA methylation [16,19,28]. We found a significant decrease in DNA methylation levels and extensive demethylation and methylation modifications after 30 days of O_3_ stress treatment, compared to the control group. Additionally, demethylation was found more frequently compared with methylation, which might be explained by more stress-related genes being activated by O_3_ stress-induced demethylation in rice genomic DNA to adapt to O_3_ stress. Therefore, these properties are expected to promote the responses and adaptation of rice to O_3_ stress; epigenetic changes in the rice genome may be a vital regulatory mechanism for the adaptation of rice to O_3_ or other environmental stresses.

## 5. Conclusions

This study showed rice’s agronomic traits effects and genome-wide DNA methylation variations in response to O_3_ stress at three different concentrations using OTC. Flag leaf length, panicle length, and hundred-grain weight showed beneficial effects at 80 nmol·mol^−1^ O_3_ and inhibitory effects at 160 and 200 nmol·mol^−1^ O_3_. The methylation-sensitive amplified polymorphism (MSAP) results showed that the O_3_-induced genome-wide methylation alterations occurred rapidly and accounted for 14.72–15.18% at three different O_3_ concentrations. Demethylation alteration was detected at the three concentrations, mainly at the CNG sites. There were 153 methylated-sensitive polymorphic sequences obtained. Interestingly, 10 stress-related recovery sequences were annotated, including hormone regulation, protein transport, plant development, and transposon activation. These results can provide a theoretical basis and a reference for more hereditary information about the molecular mechanism of plant resistance to O_3_ stress.

## Figures and Tables

**Figure 1 genes-14-01888-f001:**
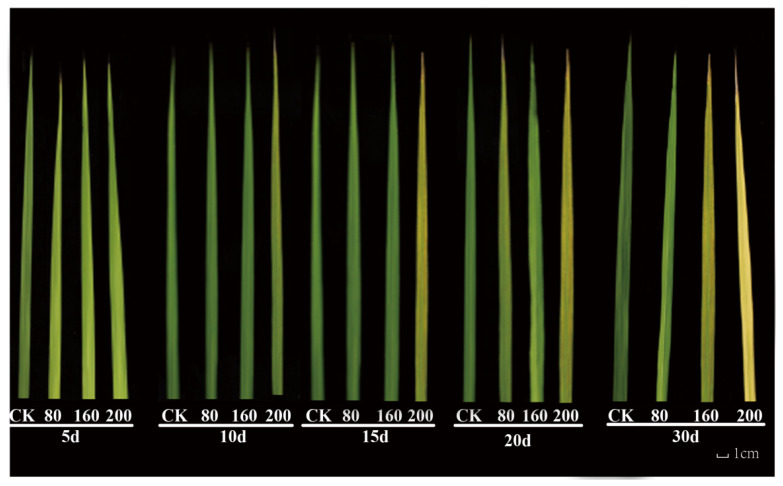
Damage to *Oryza sativa* leaves by different concentrations of O_3_ stress treatments (80 nmol-mol^−1^, 160 nmol-mol^−1^, and 200 nmol-mol^−1^).

**Figure 2 genes-14-01888-f002:**
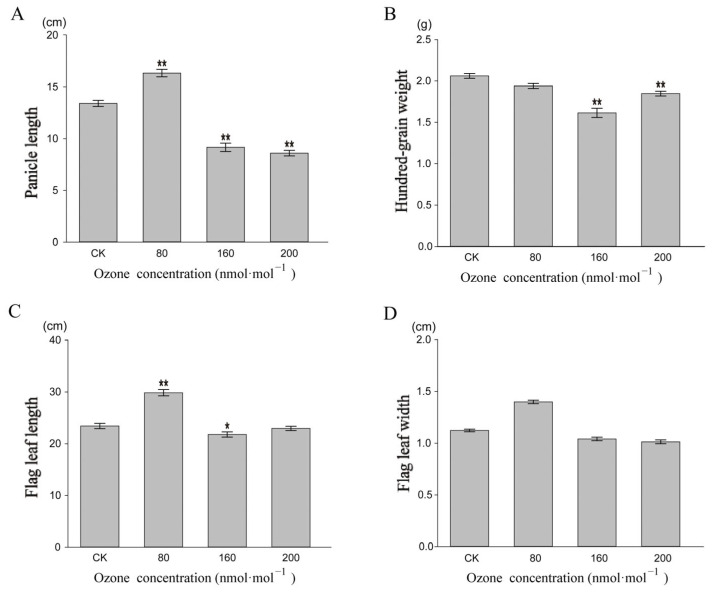
Analysis of agronomic traits of *Oryza sativa* after 30 days of O_3_ stress ((**A**): panicle length; (**B**): hundred-grain weight; (**C**): flag leaf length; (**D**): flag leaf width). * and ** denote *p* < 0.05 and *p* < 0.01, respectively.

**Figure 3 genes-14-01888-f003:**
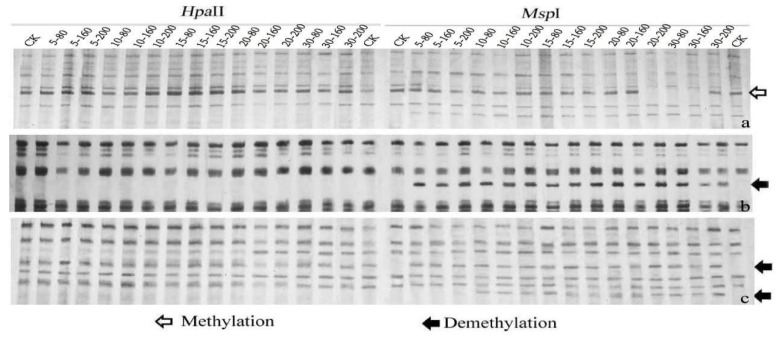
Difference of methylation patterns between samples treated with O_3_ stress at different concentrations (80 nmol-mol^−1^, 160 nmol-mol^−1^, 200 nmol-mol^−1^) (5 d, 10 d, 15 d, 20 d, 30 d) and non-treated samples (CK). Primer combinations: *Eco*RI + ACG/*Hpa*II*(Msp*I*)* + TGT (**a**), *Eco*RI + ACA/*Hpa*II*(Msp*I*)* + TTG (**b**), *Eco*RI + ATC/*Hpa*II*(Msp*I*)* + TGT (**c**).

**Figure 4 genes-14-01888-f004:**
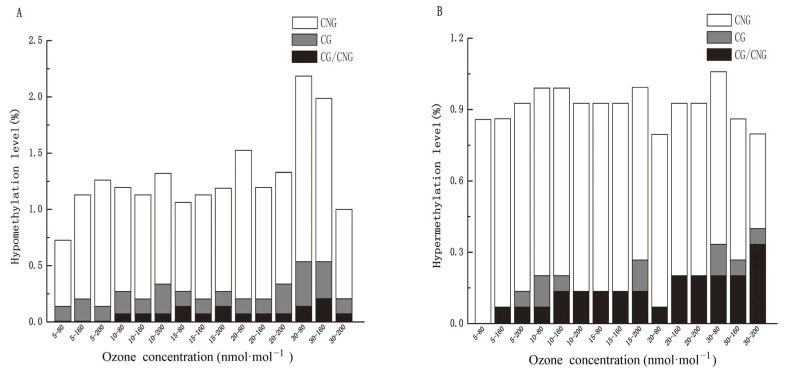
Hypomethylation (**A**) and hypermethylation (**B**) alteration in rice after O_3_ stress treatments.

**Table 1 genes-14-01888-t001:** Cytosine methylation levels at different treatments under O_3_ stress by MSAP-based method.

Type	O_3_ Stress Time-Point																
	5–CK	5–80	5–160	5–200	10–80	10–160	10–200	15–80	15–160	15–200	20–80	20–160	20–200	30–80	30–160	30–200	30–CK
I	1281	1286	1285	1286	1286	1286	1286	1290	1289	1288	1287	1288	1288	1290	1292	1292	1281
II	58	55	57	60	59	59	58	56	56	56	55	58	54	53	66	61	58
III	134	126	125	125	128	127	127	124	124	126	129	131	128	126	123	126	134
IV	42	48	48	44	42	43	44	45	46	45	44	38	45	46	34	36	42
Total sites	1515	1515	1515	1515	1515	1515	1515	1515	1515	1515	1515	1515	1515	1515	1515	1515	1515
Total amplified bands	1473	1467	1467	1471	1473	1472	1471	1470	1469	1470	1471	1477	1470	1469	1481	1479	1473
Total methylated bands	234	229	230	229	229	229	229	225	226	227	228	227	227	225	223	223	234
MSAP (%) ^a^	15.45	15.12	15.18	15.12	15.12	15.12	15.12	14.85	14.92	14.98	15.05	14.98	14.98	14.85	14.72	14.72	15.45
Fully methylated bands	176	174	173	169	170	170	171	169	170	171	173	169	173	172	157	162	176
Fully methylated ratio (%) ^b^	11.62	11.49	11.42	11.16	11.22	11.22	11.29	11.16	11.22	11.29	11.42	11.16	11.42	11.35	10.36	10.69	11.62
Hemi-methylated ratio (% ) ^c^	3.83	3.63	3.76	4.00	3.90	3.90	3.83	3.70	3.70	3.70	3.63	3.83	3.56	3.50	4.36	4.03	3.83
Non-methylated ratio (%) ^d^	84.55	84.62	84.69	84.75	84.82	84.88	84.95	85.02	85.08	85.15	85.21	85.28	85.35	85.41	85.48	85.54	84.55

^a^ MSAP (%) = [(II + III + IV)/(I + II + III + IV)] × 100. ^b^ Fully methylated ratio (%) = [(III + IV)/(I + II + III + IV)] × 100. ^c^ Hemi-methylated ratio (%) = [(II)/(I + II + III + IV)] × 100. ^d^ Non-methylated ratio (%) = [(I)/(I + II + III + IV)] × 100.

**Table 2 genes-14-01888-t002:** Chromosome location and homology analysis of variant sequences.

Fragment	Chromosome	Methylation Status	Accession	Description	E-Value	Size (bp)
		under Stress				
MS14	Chr.01	Demethylated	LOC_Os01g11054	Phosphoenolpyruvate carboxylase	3.6 × 10^−21^	86
MS93	Chr.03	Demethylated	LOC_Os03g48030	Integral membrane HPP family protein	1.7 × 10^−36^	180
MS96	Chr.04	Demethylated	LOC_Os04g40540	L-isoaspartate methyltransferase	9.3 × 10^−11^	101
MS99	Chr.11	Demethylated	XP_015616523	Purple acid phosphatase	2 × 10^−23^	177
MS104	Chr.11	Methylated	AZM68782	Hypothetical protein	1.5 × 10^−90^	291
MS107	Chr.04	Methylated	ABF98152	Transposon protein, putative, Mutator sub-class	9 × 10^−20^	164
MS111	Chr.11	Demethylated	Os11g0679400	Zinc finger family protein	4 × 10^−20^	81
MS114	Chr.06	Demethylated	LOC_Os06g43650	KH domain-containing protein	9.2 × 10^−7^	59
MS120	Chr.06	Demethylated	LOC_Os06g43650	KH domain-containing protein	1.1 × 10^−30^	133

## Data Availability

The data presented in this study are available in the article or Appendix A.

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
