# Peer review of "Growth and DNA Methylation Alteration in Rice (Oryza sativa L.) in Response to Ozone Stress"

_genes, 2023, doi:10.3390/genes14101888_

Round 1

Reviewer 1 Report

The aim of the study was to assess the response of rice to stress-induced DNA methylation changes under the influence of increased ozone response in the environment. Interesting and well-written work. Divided into subchapters that are typical for this type of article. Very nice stripes deserve positive attention, as they are often not visible during this type of test.

However, the work contains some errors that need to be corrected.

1. Lack of research hypothesis. Please complete the.

2. The term "demethylation for a given primer combination" is debatable, please read the literature and change it if necessary.

Overall, the article is interesting, and nicely written and with minor changes, it can be published in the Genes

The English language is readable and does not require any major changes.

Reviewer 2 Report

The manuscript addresses an important issue regarding the DNA methylation in rice seedlings in resposne to ozone treatments. The study is well structured, assessing the methylation the levels and patterns of cytosine methylation in rice under different concentrations of ozone stress and treatment duaration using MSAP technology. However there are a few issues that the authors should elaborate to improve the manuscript.  These are as follwo: 

1. In the Introduction, in ln 55-60 it would be nice to provide some information of the DNA methylation and specifically that of cytocine, as it is the usual nucleotide that is assessed.

2. In ln 74 O3, the number should be in subscript. The authors are ancouraged to carefuly check the manucript of minor typo errors.

3. In Materials and Methods section 2.1. the authors should describe which time duration of treatments  (i.e 5 days, 10 days, 15 or 20 days) in each ozone treatments was used to assess the agronomic traits (i.e. panicle length, flag leaf lengt, HSW). This information should be aslo shown on Figure 2.

4. The results of the above (3) should be discussed in association to the levels of the demethylation/hypermethylation taking place at the particular treatment. Also the authors should indicate why the particular time of ozone treatment was selected, providing relevant references.

Minor english typo errors should be checked.
